# Early Detrusor Application of Botulinum Toxin A Results in Reduced Bladder Hypertrophy and Fibrosis after Spinal Cord Injury in a Rodent Model

**DOI:** 10.3390/toxins14110777

**Published:** 2022-11-10

**Authors:** Juliana Y. Bushnell, Lindsay N. Cates, Jeffrey E. Hyde, Christoph P. Hofstetter, Claire C. Yang, Zin Z. Khaing

**Affiliations:** 1Department of Neurological Surgery, University of Washington, Seattle, WA 98195, USA; 2Department of Urology, University of Washington, Seattle, WA 98195, USA

**Keywords:** botulinum toxin, bladder hypertrophy, spinal cord injury

## Abstract

Following spinal cord injury (SCI), pathological reflexes develop that result in altered bladder function and sphincter dis-coordination, with accompanying changes in the detrusor. Bladder chemodenervation is known to ablate the pathological reflexes, but the resultant effects on the bladder tissue are poorly defined. In a rodent model of contusion SCI, we examined the effect of early bladder chemodenervation with botulinum toxin A (BoNT-A) on bladder histopathology and collagen deposition. Adult female Long Evans rats were given a severe contusion SCI at spinal level T9. The SCI rats immediately underwent open laparotomy and received detrusor injections of either BoNT-A (10 U/animal) or saline. At eight weeks post injury, the bladders were collected, weighed, and examined histologically. BoNT-A injected bladders of SCI rats (SCI + BoNT-A) weighed significantly less than saline injected bladders of SCI rats (SCI + saline) (241 ± 25 mg vs. 183 ± 42 mg; *p* < 0.05). Histological analyses showed that SCI resulted in significantly thicker bladder walls due to detrusor hypertrophy and fibrosis compared to bladders from uninjured animals (339 ± 89.0 μm vs. 193 ± 47.9 μm; *p* < 0.0001). SCI + BoNT-A animals had significantly thinner bladder walls compared to SCI + saline animals (202 ± 55.4 μm vs. 339 ± 89.0 μm; *p* < 0.0001). SCI + BoNT-A animals had collagen organization in the bladder walls similar to that of uninjured animals. Detrusor chemodenervation soon after SCI appears to preserve bladder tissue integrity by reducing the development of detrusor fibrosis and hypertrophy associated with SCI.

## 1. Introduction

Approximately 80% of individuals who suffer a traumatic spinal cord injury (SCI) will develop neurogenic bladders and the many associated complications. Neurogenic bladder is a general term used to describe bladder malfunction related to nervous system injury or disease. The bladder and the nervous system are intimately synchronized to allow for healthy urine storage and release [1]. The coordinated contraction of the detrusor and relaxation of the sphincters can be impaired in patients with neurogenic bladders, particularly those due to SCI. Following suprasacral SCI, detrusor hyperreflexia and detrusor sphincter dyssynergia occur resulting in high pressures within the bladder [2,3]. This results in detrusor hypertrophy, and collagen deposition [4,5,6], that in turn diminishes bladder compliance [7], causing a multitude of problems including urinary incontinence, urinary tract infection and renal failure.

Currently, botulinum toxin type A (BoNT-A) is approved for clinical use for treating symptoms of neurogenic bladder, including urinary frequency and incontinence, but only when other treatments such as oral anticholinergic treatments fail. BoNT-A is known to assert its actions by directly blocking the efferent cholinergic neurotransmission directly, but it also appears to inhibit the excitatory neurotransmission from the afferents and uroepithelium [8,9]. Chemodenervation by BoNT-A can reduce symptoms due to detrusor overactivity, and improve bladder compliance [10]. However, since BoNT-A injections are considered only after other treatment options fail, which can be months to years after the initial onset of symptoms, during that time, hypertrophy and fibrosis of the bladder could have already begun [5], causing permanent tissue changes that contribute to bladder dysfunction. In this report, we present findings on the early application of BoNT-A into the detrusor in a rodent model of SCI. Our hypothesis is that early bladder chemodenervation can prevent or delay the onset of the inimical bladder wall changes after spinal injury.

## 2. Results

### 2.1. General Animal Health

One animal from each of the SCI + saline and SCI + BoNT-A groups was euthanized prior to the scheduled endpoint, due to the animal reopening the abdominal sutures after laparotomy. Their tissue and data were excluded from this study. To standardize the degree of injury, we excluded any animal whose force was 20% from the mean force during the contusion injury, as measured by the IH device. One animal from SC + saline group was excluded, as it did not reach our injury force inclusion criteria.

There were no body weight differences between SCI + saline, and SCI + BoNT-A animals (276 ± 7 g vs. 277 ± 7 g; *p* > 0.05).

### 2.2. Bladder Weight

The bladders of SCI + saline animals were significantly heavier than spinal intact animals (241 ± 25 mg vs. 137 ± 15 mg; *p* < 0.01). SCI + saline bladders were also significantly heavier compared to the bladders of spinal cord injured animals with intradetrusor BoNT-A injections (SCI + BoNT-A) (241 ± 25 mg vs. 183 ± 39 mg; *p* < 0.05) (Figure 1).

### 2.3. Histological Changes

SCI + saline animals had significantly thicker bladder walls compared to spinal intact animals (339 ± 89.0 μm vs. 193 ± 47.9 μm; *p* < 0.0001) (Figure 2). H and E stains showed the development of detrusor hypertrophy and hyperplasia after SCI. Bladders of the SCI + BoNT-A group had significantly thinner bladder walls compared to the bladders of SCI + saline animals (202 ± 55.4 μm vs. 339 ± 89.0 μm; *p* < 0.0001) and had similar thickness to that of spinal intact animals (193 ± 47.9 μm vs. 202 ± 55.4 μm; *p* > 0.05). Moreover, limited hyperplasia of the urothelium was also noted in SC + BoNT-A treated animals compared to SCI + saline animals (Figure 2). We also noted urothelial desquamation in all SCI + saline rats, but not in BoNT-A treated animals.

### 2.4. Collagen Directionality

Microscopic analysis of the Masson’s trichrome staining of the bladder section revealed visible differences in collagen deposition (blue staining) and orientation between SCI + saline bladders and SCI + BoNT-A bladders (Figure 3). Collagen within the SCI + BoNT-A bladders retained the smooth and aligned nature of the collagen fibers seen in the bladders of uninjured animals, whereas SCI + saline injected bladders had much thicker and more disorganized collagen deposits, suggesting increased collagen deposition, although this was not directly measured.

Directionality analysis of the bladder sections stained with Mason’s trichrome staining showed that collagen fibrils of spinal intact animals consistently produced histograms with defined peaks (centered around 0 degree; Figure 3). Histograms generated from SCI + saline bladders showed no discernable peak or pattern, indicating a more random distribution of collagen directionality (Figure 3). SCI + BoNT-A injected bladders resulted in better-defined bell curve histograms, similar to those of bladders from spinal intact animals.

### 2.5. Bladder Function after SCI

Based on mean expressed bladder volumes, bladder reflexes returned more quickly in the group treated with BoNT-A than the saline treated group. SCI + BoNT-A group had lower residual bladder volumes than the SCI + saline group, with a statistically significant difference detected at 4 dpi (0.95 ± 0.31 mL vs. 2.07 ± 0.14 mL; *p* < 0.001) (Figure 4). By 8 dpi, the mean bladder volume for SCI + BoNT-A group was 0 mL during manual bladder expressions in the AM (i.e., the bladders were mostly empty) whereas SCI + Saline was 0.4 mL. These results indicate that early denervation improves functional outcomes post SCI.

### 2.6. Spinal Cord Injury and Motor Function

The SCI + saline and SCI + BoNT-A groups both experienced similar degrees of locomotor impairments at all time points examined on the BBB scale, confirming that injury severity was the same between the two SCI groups (Figure 5A). Lesion size analysis of the injured spinal cord (Figure 5B) confirmed that the average percentage of intact spinal tissue was comparable between the SCI + saline group and the SCI + BoNT-A group (86.16 ± 0.86% vs. 86.25 ± 0.96%; *p* > 0.05) (Figure 5C).

## 3. Discussion

The results of this study demonstrate that in rodents, it is possible to prevent or delay the onset of deleterious tissue changes associated with SCI, with early detrusor chemodenervation. Normal bladder weight and wall thickness, and collagen deposition and organization were maintained following SCI, with early chemodenervation.

Previously, Temeltas et al. examined intradetrusor injections of BoNT-A after SCI in a rodent model and found that both early (7 dpi) and late (28 dpi) BoNT-A injections resulted in significant reductions in fibrosis and hypertrophy as well as bladder pressure [6]. However, they found no significant differences in the tissue changes detected with histology between “early” and “late” application of BoNT-A. One explanation can be found in the study by Zinck et al., who noted that by 8 dpi, the morphological and tissue changes to the rat bladder following SCI, such as hypertrophy and fibrosis, had already developed, but not before 3 dpi in rats [11]. This time-course of hypertrophic and fibrotic bladder development suggests that BoNT-A injections at 7 dpi might be too late to inhibit the development of the initial bladder tissue changes. The application of BoNT-A immediately after SCI in this study appeared to prevent much of the bladder tissue changes after SCI in rats. Interestingly, a recent clinical study examining detrusor function during the acute SCI period in patients revealed that almost two-thirds showed unfavorable urodynamic parameters within the first 40 days after SCI [12]. The data from the current study suggest that early intervention to limit neurogenic bladder dysfunction during the acute phase after SCI could improve the long-term bladder outcomes for SCI patients. Future studies will need to determine the full therapeutic window for BoNT-A treatment to achieve tissue preservation.

In addition to timing of the BoNT-A application, dosing, and distribution of the denervating agent still needs to be worked out for optimal outcomes. A recent study showed that BoNT-A 7.5 U injected at eight spots in the detrusor, sparing the trigone, is an optimal dose for inhibiting the effects of neurogenic detrusor overactivity after spinal transection in Sprague Dawley rats. However, this study did not examine bladder tissue changes after BoNT-A injection; only functional cystometry studies were performed [13]. Temeltas et al., examined the effects of detrusor BoNT-A injections on bladder tissue but used only 2–3 U of BoNT-A per injection site, with 10–12 injection sites resulting in 20–30 U of BoNT-A delivered [6]. Yet another study examined the number of injection sites (four vs. eight) of BoNT-A (total dose = 22.5 U) into the detrusor muscle and found that four injection sites was just as effective at increasing the bladder capacity and compliance compared to eight site injections [14]. In the current study, we chose to use 10 U of BoNT-A distributed over eight injection sites into the detrusor muscle. We also chose to spare the trigone, as injecting the trigone would have required a cystotomy, introducing another variable (tissue trauma) into the analysis.

Connective tissue fibril alignment can play a significant role in the mechanical properties of the bladder tissue [15], and increased collagen deposition within the bladder walls has been associated with non-compliant bladder function SCI in rodent models [16,17]. Here, we showed that acute chemodenervation of the bladder with BoNT-A resulted in reduced collagen deposition and preserved collagen alignment within the lamina propria of the bladder, consistent with a more compliant bladder.

All post-SCI animals went through manual bladder expression until their voiding reflexes returned (~10–14 dpi). While this is an efficient and effective method of relieving urine built up in the bladder, there are concerns that this type of manual manipulation can result in bladder tissue damage. However, a recent study showed that increasing the number of manual bladder expression can improve the storage and voiding bladder dysfunction associated SCI in mice [18]. In addition, all post SCI animals went through a similar number of manual bladder expressions (twice a day), and therefore comparison of bladder tissue thickness between different treatment groups is still valid.

### Translational Relevance and Study Limitations

The mechanism of action was not defined in this study, but presumably is due to the ablation of pathologic bladder reflexes arising following spinal injury. The reflex ablation could occur by damaging the afferent signals of bladder pressure or distension or preventing the efferent signals for detrusor contraction. Either mechanism, or a combination of the two, may be occurring.

In humans, suprasacral SCI results in development of pathologic bladder reflexes: uninhibited contractions and detrusor sphincter dyssynergia. These pathologic reflexes tax the detrusor, and, along with other non-neurogenic factors, result in development of detrusor tissue changes: hypertrophy, fibrosis, and collagen deposition. The normally compliant detrusor becomes less compliant, and its filling capacity is diminished. Poor compliance contributes to exacerbated incontinence and upper tract complications such as hydronephrosis, pyelonephritis, and loss of renal function [19].

The action of chemodenervation mimics the condition of patients who have sacral SCIs, who develop different neurogenic bladder pathologies than patients with more rostral injuries. Specifically, lower spinal injuries result in a non-contracting and low-pressure bladder—a “lower motor neuron” organ. The loss of contractility, in terms of frequency, strength, and duration of bladder contractions, avoids the development of fibrous and hypertrophic bladder tissue found in patients with SCI at higher spinal segments. The peripherally denervated bladder, while unable to function to evacuate urine, does preserve its role as a low-pressure reservoir. Low-pressure urine storage is necessary for the social function of continence. But more importantly, it is essential to minimize upper urinary tract complications of the neurogenic bladder, which are much more morbid than those of the lower urinary tract.

Our study was limited to histological analysis, and in future studies functional cystometry is necessary to examine the effects of acute BoNT-A treatment on bladder compliance and function. Another limitation is the inability to define the subcellular mechanisms of tissue preservation. Examining the mechanism of how chemodenervation by BoNT-A can limit bladder hypertrophy and fibrosis after SCI will allow this method to be further exploited for preserving bladder health. Several studies have shown that increased expression of nerve growth factor (NGF) in the bladder is associated with bladder hypertrophy and dysfunction in both pre-clinical model of SCI [20,21] and in patients [22,23].

## 4. Conclusions

Bladder chemodenervation with BoNT-A early after SCI reduced the development of pathologic detrusor histology by decreasing hypertrophy, fibrosis, and disorganized collagen deposition and in a rodent SCI contusion model. These data point to the potential of early use of intradetrusor injection of BoNT-A for patients with SCI, as a prophylaxis against the most severe complications of neurogenic bladder.

## 5. Materials and Methods

Three groups of rats were investigated in this study. An uninjured control group (sham; N = 5) underwent laminectomy only. The remaining two groups underwent T9 severe contusion injuries following laminectomy and either received saline (SCI + saline) (N = 8) or BoNT-A injections (SCI + BoNT-A) (N = 8) into the detrusor. Animals were then monitored twice daily for up to two weeks post injury (wpi) or until bladder reflexes returned. At 8 wpi, animals were sacrificed, and bladders were collected, weighed, and processed for histological analysis.

### 5.1. Spinal Cord Injury Model

Sterile surgeries were performed on female Long-Evans rats (between 250–350 g; Harlan Laboratories) per the Institutional Animal Care and Use Committee (IACUC) protocol and the Office of Laboratory Animal Welfare (OLAW) guidance. Isoflurane (3–5% to induce, 1.5–3% to maintain) was used as general anesthesia, and the skin at the incision site was injected with 0.2 mL of 0.25% bupivacaine as a local anesthetic. After sterile preparation, a laminectomy was performed at T8/9 to expose the spinal cord. The sham animals were closed without spinal injury and allowed to recover. Animals in the two injury groups received a severe contusion at 200-kdyne with zero dwell time using the Infinite Horizon device (Precision Systems and Instrumentation, Fairfax Station VA). This contusion type SCI model has been well established with well-characterized histological and functional changes in the bladder [24,25,26],

Every 8–12 h for 48 h after injury, the animals were injected with buprenorphine (0.03 mg/kg) subcutaneously to manage the pain associated with the surgical procedure. Bladders were expressed manually two to three times a day; urine volume and bladder sizes were recorded until spontaneous voiding reflexes returned (about 2 wpi). Antibiotics were administered daily for 5 days post injury (dpi) as infection prevention (gentamicin, 5 mg/kg Pfizer, Inc.).

### 5.2. Intradetrusor Injection

Before undergoing surgery, bladders were manually expressed. After an appropriate level of anesthesia was obtained, a laparotomy was performed to expose the bladder. The dome of the bladder was carefully lifted from the surrounding abdominal tissue using a sterile plastic loop. Eight micro-injections of either saline (N = 8) or BoNT-A (onabotulinumtoxinA, Botox^®^, Allergan, Inc., Madison, NJ, USA) (N = 8) were made into the detrusor using a 33-gauge needle attached to a Hamilton syringe (Model 750 N Syringe, Hamilton Company, Reno, Nevada) and an infusion pump (Single Syringe Infusion Pump, Stoelting Company, Wood Dale, IL, USA) at a rate of 1 microliter/min with either sterile saline or 10 U of BoNT-A suspended in 24 microliters saline (3 microliters/injection). The needle was left in place for 3 min after each injection to limit flow back. Both saline and BoNT-A solutions were tinted with small amounts of sterile methylene blue (1% solution) to visually confirm successful injections. The abdominal incision was then sutured closed, and animals were maintained for 8 wpi.

### 5.3. Bladder Volumes

Residual bladder volumes were measured as a means of indicating the return of bladder reflexes following SCI. Bladders were manually expressed twice daily until autonomous voiding was restored (up to 14 dpi). The amount of urine expressed was judged visually and converted as urine volume (empty~0.05, very small ~0.5 mL, small ~1.5 mL, medium ~3.5 mL or large ~4.5 mL). This scale was developed by weighing the urine after daily manual expression in a previous study in our lab (*n* = 53 rats). The measurements were all performed by the same two experimenters.

### 5.4. Locomotion after SCI

The Basso, Beattie, and Bresnahan (BBB) locomotor scale is a standard test to grade the locomotor recovery of chronic SCI rats [27]. Rats are scored according to a standardized score sheets from 0 (no hindlimb movement) to 21 (complete locomotor recovery). Rats are placed in an open-field arena for 4 min once a week post injury. Two trained scorers, blinded to the injury and treatment status, assign a score to each animal by observing for joint movement, stepping, and coordination according to the prescribed categories. Video recordings of the rats’ activity were also made but used to review only when there was a discrepancy between the scorers. Open-field locomotor behavioral test (BBB) was conducted at least one week before injury and weekly thereafter in all rats.

### 5.5. Histological Analysis

Animals were sacrificed at 8 wpi with an overdose of pentobarbital sodium (intraperitoneal, 0.39 g/mL Schering-Plough Animal Health, Madison, NJ, USA). All bladders were manually expressed prior to sacrifice. Animals were fixed via trans-cardiac perfusion with 200 mL of ice-cold PBS (pH 7.4) followed with 200 mL of 4% paraformaldehyde. Bladders were then harvested and stored in 4% paraformaldehyde at 4 °C for 24 h. Three blocks were made from each bladder (dome, middle and base) and three slices from each part of the bladder (dome, middle, and base) were mounted onto slides in 5μm sections. Standard hematoxylin and eosin and Masson’s trichrome staining were conducted on the sections.

Entire spinal columns were also removed, post-fixed in 4% paraformaldehyde solution overnight at 4 °C and treated with 30% sucrose solution with 0.01% sodium azide prior to freezing. Seven-millimeter segments centered over T6–10 was dissected, and frozen, and 20-μm-thick axial sections were obtained using a cryostat (Leica, CM1850), thaw mounted onto gelatin-coated glass slides, and stored at −80 °C. Standard cresyl violet and myelin staining was performed on axial sections to determine extent of SCI.

Histology slides were imaged using a Zeiss Primo Star microscope with a color camera (Axiocam ERc 5 s). Multiple measurements were taken of processed images using ImageJ [28] such as bladder circumference, width, muscle fiber diameter, and epithelial thickening using the straight- and freehand-line tool. Collagen directionality was measured using the “Directionality” plugin available on ImageJ. After selecting a 250 × 250-pixel area of collagen and creating a new black and white image, the image was processed using the “find edges” tool. Next, the edge image was analyzed using the directionality tool, which produced a directionality histogram.

Five spinal cord sections within 1 mm of the injury center were also selected per animal for analysis. The percentage of tissue spared was analyzed using the freehand line tool.

### 5.6. Statistical Analysis

All statistical analyses were performed using GraphPad Prism 7.0 for Mac (San Diego, CA, USA). Histological and functional data were analyzed using 2-way analysis of variance (ANOVA) with Tukey’s test to determine significance of BoNT-A use. Bladder weight data were analyzed using one-way ANOVA with post hoc Bonferroni.

## Figures and Tables

**Figure 1 toxins-14-00777-f001:**
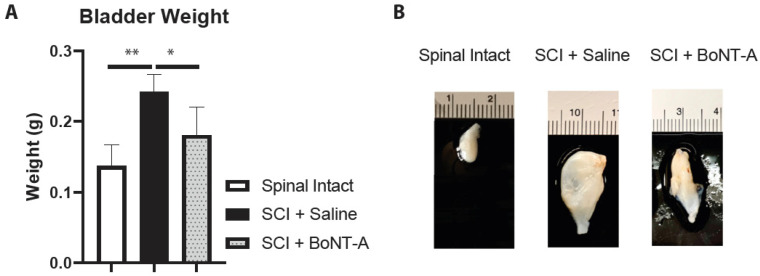
Acute detrusor BoNT-A injections after SCI resulted in lighter bladders. (**A**) Detrusor BoNT-A injections results in significantly lighter bladders after SCI. (* *p* < 0.05, ** *p* < 0.01). (**B**) Representative photos bladders from the spinal intact and experimental groups showing visible differences between bladder sizes.

**Figure 2 toxins-14-00777-f002:**
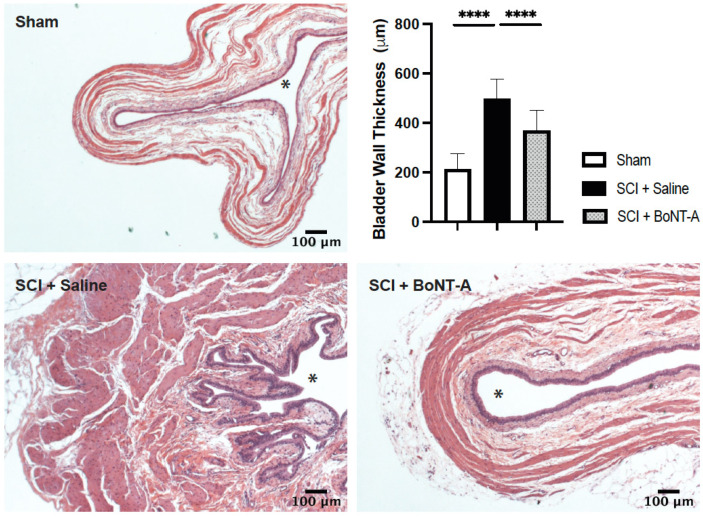
Acute detrusor BoNT-A injections after SCI resulted in reduced fibrosis and bladder wall thickening. Representative photomicrographs of Hematoxylin and Eosin (H&E) staining of bladder sections from spinal intact Sham (*n* = 5), SCI + Saline (*n* = 8) and SCI + BoNT-A (*n*= 8) groups at 8 wpi. Asterisk (*) in histology images denotes bladder lumen. SCI_+_ Saline group had significantly thicker bladder walls compared to spinal intact animals (****, *p* < 0.0001). SCI + BoNT-A group had significantly thinner bladders compared to the bladders of SCI + Saline animals (****, *p* < 0.0001). Scale bar = 100 micrometer.

**Figure 3 toxins-14-00777-f003:**
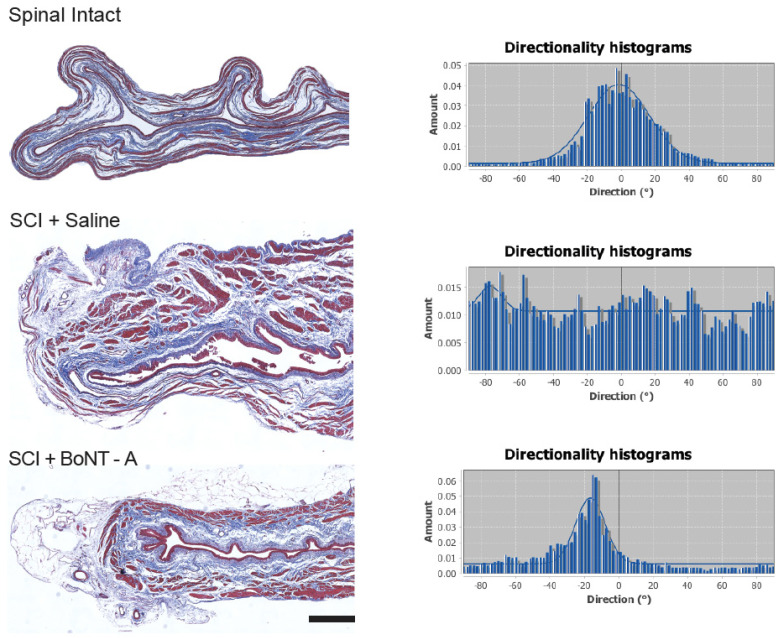
Acute detrusor BoNT-A injections after SCI resulted in ordered collagen directionality in the detrusor. Mason Trichrome staining showed keratin/muscle fibers in red, and collagen in blue. The left column shows representative photomicrographs from each of the experimental and control groups. The right column shows directionality histograms that are standard for their corresponding group. Relatively organized collagen has a single defined peak whereas unorganized collagen has no distinct peak. Directionality of collagen deposition was disorganized in the bladders of SCI + Saline animals (row 2). In the bladders of SCI + BoNT-A animals (row 3), the histogram showed a preferred orientation and organization similar to that of spinal intact animals (row 1). Scale bar = 100 micrometer.

**Figure 4 toxins-14-00777-f004:**
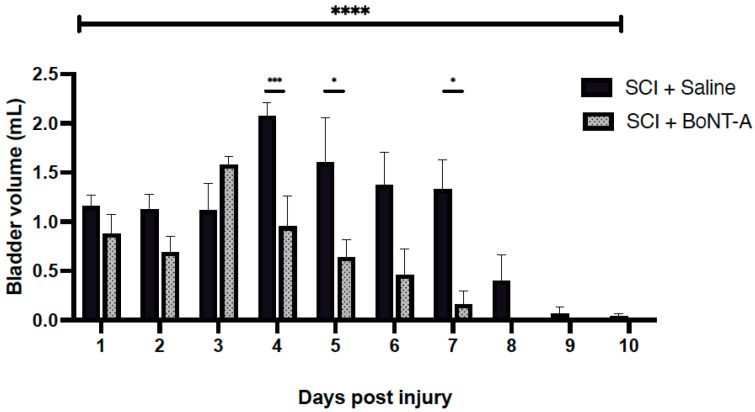
Acute detrusor BoNT-A injections after SCI resulted in faster return of voluntary micturition. Bladder volumes were recorded daily in the AM for 10 days post injury. The SCI + Saline group (*n* = 8) had significantly higher bladder volume than the SCI + BoNT-A group (*n* = 8 **** *p* < 0.0001). *** *p* < 0.001, * *p* < 0.05.

**Figure 5 toxins-14-00777-f005:**
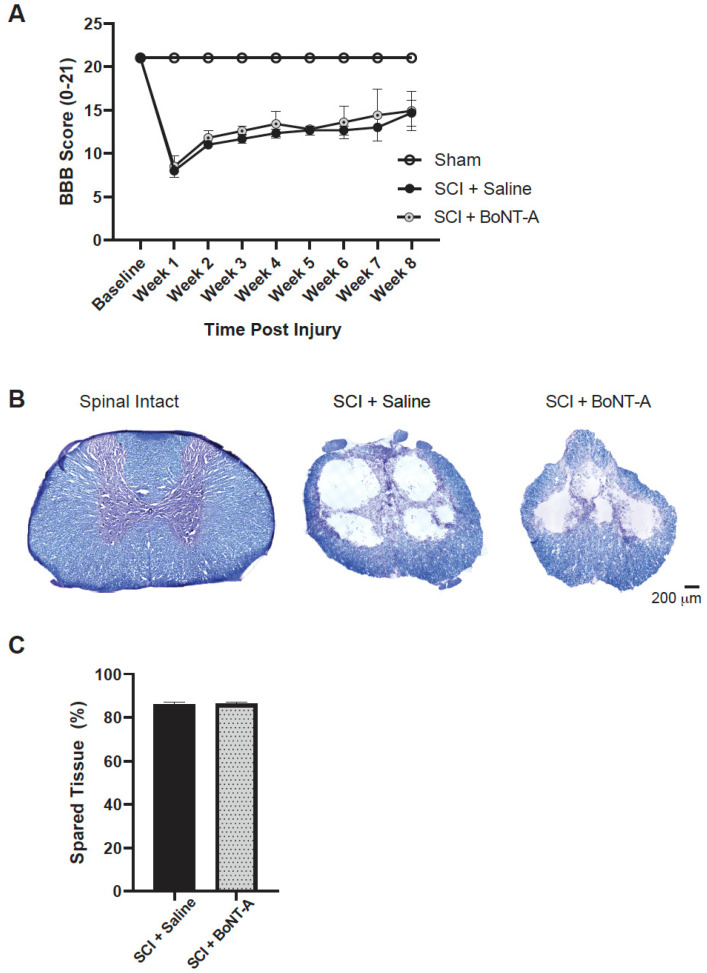
Functional deficit and lesion size was similar between SCI + Saline and SCI + BoNT-A groups: (**A**) weekly BBB scores showed that there were no significant behavioral differences in the locomotor recovery for up to 8 wpi between the SCI + Saline group and the SCI + BoNT-A group (*p* > 0.05); (**B**) Representative photomicrographs of Nissl stained 20 μm thick sections of spinal intact, SCI + Saline, and SCI + BoNT-A spinal cords are shown. Scale bar = 200 μm; and (**C**) Nissl and myelin-stained spinal cord sections within 1 mm of the injury center were analyzed for percentage of spared tissue. There were no statistical differences between the SCI + Saline and SCI + BoNT-A group (86.16 ± 0.86% vs. 86.25 ± 0.96%; *p* > 0.05).

## Data Availability

Not applicable.

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
