# Peer review of "Early Detrusor Application of Botulinum Toxin A Results in Reduced Bladder Hypertrophy and Fibrosis after Spinal Cord Injury in a Rodent Model"

_toxins, 2022, doi:10.3390/toxins14110777_

Round 1

Reviewer 1 Report

This is a very well conducted experiment and beautifully written manuscript. Methods are comprehensively described, with the single exception of the Bladder volumes paragraph: "The amount of urine expressed was judged visually and recorded". Isn't there a more objective way of measuring expressed bladder contents? Were all 'measurements' performed by the same author?  Is bladder expression twice daily sufficient and representative ot the normal voiding pattern of animals? Please dicuss the appropriateness and any potential bias which might occur by such methodology. 

Author Response

Comment 1: This is a very well conducted experiment and beautifully written manuscript.

Response 1: We thank the reviewer for their kind words.

Comment 2: Methods are comprehensively described, with the single exception of the Bladder volumes paragraph: "The amount of urine expressed was judged visually and recorded". Isn't there a more objective way of measuring expressed bladder contents? Were all 'measurements' performed by the same author? 

Response 2:  Because the rat’s bladder empties incompletely following SCI, the urine must be manually expressed by pressing on the lower abdomen. This is a standard way to perform and report this maneuver. Additionally, our lab has developed a scale for urine volume based on the bladder size. This scale was developed by weighing the urine after daily manual expression (n =53 rats).  The bladder sizes were categorized as very small, small, medium, and large and the average measured volume from each category were 0.04, 0.33, 1.70, 3.25, 4.45 mL respectively. These numbers were then used to convert bladder size to a volume measurement. This information is now added to the method’s section. All the measurements were performed by the same two researchers throughout the development of the scale. The measurements were all performed by the same author.  The following has been added to the Methods section to clarify this.  

“This scale was developed by weighing the urine after daily manual expression in a previous study in our lab (n =53 rats). The measurements were all performed by the same two experimenters.”

Comment 3: Is bladder expression twice daily sufficient and representative of the normal voiding pattern of animals?

Response 3: Yes. Bladder expression twice daily in rats with thoracic contusions are sufficient for the health of the animals with almost no complications. However, this type of manual bladder expression is not likely to represent normal voiding patterns of the animals.

Comment 4: Please discuss the appropriateness and any potential bias which might occur by such methodology (manual bladder expression).

Response 4: We agree with the reviewer and the following has been added to the discussion.

“Post-SCI animals went through manual bladder expression until their voiding reflexes return (~ 10 – 14 dpi). While this is an efficient and effective method of relieving urine built up in the bladder, there are concerns that this type of manual manipulation can result in bladder tissue damage. However, a recent study showed that increasing the number of manual bladder expression can improve the storage and voiding bladder dysfunction associated SCI in mice (Wada, Shimizu et al. 2017). In addition, all post SCI animals went through a similar number of manual bladder expressions (daily twice a day), and therefore comparison of bladder tissue thickness between different treatment groups is still valid. “

Reviewer 2 Report

The authors present an interesting article entitled“Early Detrusor Application of Botulinum Toxin A Results in Reduced Bladder Hypertrophy and Fibrosis after Spinal Cord Injury in a Rodent Model. “

The introduction of the article is well-designed for the purpose of the manuscript. Materials and Methods are clearly and comprehensively presented. Statistical analysis is correct for the purpose.

The discussion section can be shortened by avoiding redundancies because the Results section is already well presented (lines 151-158). I suggest that the names of the authors in the text should be avoided as much as possible (the authors are clearly presented with citations in bracelets at the end of the sentence). The figures are correct and illustrative. I suggest that Figure 5 is better suited as Supplementary material as it deals with no significant behavioral differences in the locomotor recovery between functional deficit and lesion size between the two groups. Section 3.1 heading should be revised to contain limitations of the study clearly pointed. I suggest limitations of the study and possible future research. 

Author Response

Comment 1: The discussion section can be shortened by avoiding redundancies because the Results section is already well presented (lines 151-158).

Response 1: We agree with the reviewers and have shortened the description of results in the discussion section.

Comment 2: I suggest that the names of the authors in the text should be avoided as much as possible (the authors are clearly presented with citations in bracelets at the end of the sentence)

Response 2: Thank you. We agree with the reviewer and have made edits throughout the manuscript.

Comment 3: I suggest that Figure 5 is better suited as Supplementary material as it deals with no significant behavioral differences in the locomotor recovery between functional deficit and lesion size between the two groups.

Response 3: We feel strongly that this data remains in the main results section.

Comment 4: Section 3.1 heading should be revised to contain limitations of the study clearly pointed. I suggest limitations of the study and possible future research. 

Response 4: We agree and have made edits to the head of section 3.1.